# Study protocol for evaluation of aid to diagnosis for developmental dysplasia of the hip in general practice: controlled trial randomised by practice

Andreas Roposch ,[1,2] Kaltuun Warsame,[1] Angel Chater,[3] Judith Green,[4] Rachael Hunter ,[5] John Wood,[6] Nick Freemantle,[7] Irwin Nazareth[5,6]

¹Great Ormond Street Institute of Child Health, UCL, London, UK
²Department of Orthopaedic Surgery, Great Ormond Street Hospital for Children, London, UK
³Department of Sport Science and Physical, University of Bedfordshire, Luton, UK
⁴Department of Population Health Sciences, Kings College London, London, UK
⁵Research Department of Primary Care and Population Health, UCL, London, UK
⁶PRIMENT Clinical Trials Unit, UCL, London, UK
⁷Comprehensive Clinical Trials Unit, UCL, London, UK

**Correspondence to**
Professor Andreas Roposch;
a.roposch@ucl.ac.uk

## ABSTRACT

**Introduction** In the UK, a compulsory '6-week hip check' is performed in primary care for the detection of developmental dysplasia of the hip (DDH). However, missed diagnoses and infants incorrectly labelled with DDH remain a problem, potentially leading to adverse consequences for infants, their families and the National Health Service. National policy states that infants should be referred to hospital if the 6-week check suggests DDH, though there is no available tool to aid examination or offer guidelines for referral. We developed standardised diagnostic criteria for DDH, based on international Delphi consensus, and a 9-item checklist that has the potential to enable non-experts to diagnose DDH in a manner approaching that of experts.

**Methods and analysis** We will conduct a controlled trial randomised by practice that will compare a diagnostic aid against standard care for the hip check. The primary objective is to determine whether an aid to the diagnosis of DDH reduces the number of clinically insignificant referrals from primary care to hospital and the number of late diagnosed DDH. The trial will include a qualitative process evaluation, an assessment of professional behavioural change and a full health economic evaluation. We will recruit 152 general practitioner practices in England. These will be randomised to conduct the hip checks with use of the study diagnostic aid and/or as per usual practice. The total number of infants seen during a 15-month recruitment period will be 110 per practice. Two years after the 6-week hip check, we will measure the number of referred infants that are (1) clinically insignificant for DDH and (2) those that constitute appropriate referrals.

**Ethics and dissemination** This study has approval from the Health Research Authority (16/1/2020) and the Confidentiality Advisory Group (18/2/2020). Results will be published in peer-reviewed academic journals, disseminated to patient organisations and the media.

**Trial registration number** NCT04101903; Pre-results.

## INTRODUCTION

Developmental dysplasia of the hip (DDH) is characterised by varying displacement of the proximal femur from the acetabulum with associated acetabular dysplasia. Dislocation occurs in 1–2/1000 infants per year but

### Strengths and limitations of this study

► To our knowledge, this is the first trial to evaluate a diagnostic aid for the 6-week check with reference to evaluating both missed and unnecessary referrals to hospital.
► Implementable on existing clinical software used by general practitioners, the proposed aid will be easy to use.
► A comprehensive process evaluation, using qualitative methods and behavioural change frameworks, will be done alongside to the trial; plus a full health economic evaluation.
► The reliance on existing practice staff to report monthly updates is essential but could pose a risk to timely and complete data collection.
► The collection of identifiable data through site visits across 152 practices will be challenging.

milder forms occur in 40–60/1000.[1] Early recognition of disease is associated with better outcomes. It is national policy[2] to examine all infants for the presence of DDH at birth and at age 6–8 weeks in primary care (6-week hip check). If diagnosed within the first 6–8 weeks, splinting of the hips is successful in 85% of cases.[3] Later diagnoses usually will require invasive treatment, with many years of continued monitoring.[4] Late diagnosed DDH is a common cause of medical negligence claims, with increased suffering for affected patients.

Timely diagnoses of DDH remain a challenge despite the compulsory 6-week check.[5] In one study, the median age at diagnosis was 14 months, with only 40% of infants diagnosed during routine examinations and 60% presenting owing to parental concerns.[6] In another study, 30% of infants were not diagnosed by 12 weeks.[7] Because there is no further compulsory check after that at 6 weeks, it is vital that the 6-week check is effective.

Also infants incorrectly identified with 'DDH' in primary care and referred to hospital remain a challenge for both families and National Health Service (NHS). This group does not require treatment and reassurance provided in primary care would avoid unnecessary anxiety and inefficient use of hospital resources. Of 1918 infants referred to hospital from primary care for DDH, only 64 (3%) had DDH[7] but 1270 (66%) were identified as 'DDH' based on inappropriate criteria,[8] for example, 'crease asymmetry' (n=234) or 'click' (n=648). If general practitioners (GPs) were able to discriminate better between benign abnormalities requiring reassurance (perhaps with a follow-up in primary care) and findings requiring referrals, outcomes of infants with and without DDH would improve.

In prior research we suggested that too many diagnostic criteria[8] and variability in the use of diagnostic criteria among clinicians[9] complicate the task of diagnosing DDH. We developed standardised diagnostic criteria to reduce the variability in assessment and management decisions in infants examined for DDH.[10] The weighted criteria demonstrated validity.[10] In a study of 44 patients referred from GPs to hospital, the weighted criteria demonstrated a positive predictive value of 89% (95% CI 70% to 97%) and negative predictive value of 76% (95% CI 50% to 96%).[11] We refined these binary criteria, in form of a checklist with nine items, for use by GPs during the 6-week hip check and developed a training video, featuring a GP. This checklist and video were refined in a feasibility study which used qualitative methods to explore the acceptability of the format, style and delivery of the intervention (report available on request). The video explains the meaning of the diagnostic criteria; it also demonstrates how precisely to elicit the diagnostic criteria. For example, the video explains the difference between the Barlow and Ortolani manoeuvres, how to test for a leg length inequality or how to identify limitations in hip abduction.

### Rationale

Current referral patterns suggest there could be considerable health gains from improved diagnostic and referral decisions at the 6-week check.[5] GPs, in a preparatory focus group, identified the need for a 'comprehensive, structured guide' for the 6-week check. Diagnostic aids enable physicians to overcome barriers in diagnostic reasoning[12] by shifting intuitive to analytical aspects of diagnostic reasoning. Decisions made under these circumstances approach normative reasoning and rationality more closely, and are more reliable and safer.[13] Building on our earlier research, we propose to facilitate GP's diagnostic reasoning at the 6-week check by structuring current practice with use of a previously developed diagnostic aid. A diagnostic aid of this kind provides a structured approach to the assessment of infants and offers guidance about referrals.

### Objectives

► To determine whether an aid to the diagnosis of DDH reduces the number of clinically insignificant referrals

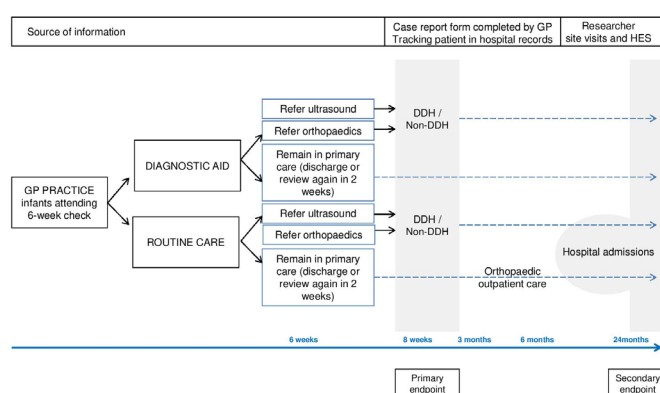

**Figure 1** Flow chart demonstrating pathways and principal endpoints and their collection. DDH, developmental dysplasia of the hip; GP, general practitioner; HES, Health Episodes Statistic.

from primary care to hospital, and the number of late diagnosed DDH.
► To determine the cost-effectiveness of this intervention.
► To conduct an integrated qualitative and quantitative process evaluation in order to understand all participants' experience in the trial and with the intervention; study how the intervention is implemented; investigate contextual factors that affect the intervention.

## METHODS AND ANALYSIS

The Standard Protocol Items: Recommendations for Interventional Trials reporting guidelines were used in the preparation of this clinical trial protocol.[14]

### Design

This is a cluster randomised controlled trial of GP practices in England, with randomisation and intervention at practice level but the primary endpoint measured at patient level (figure 1). An internal pilot will be done in the first 4 months to ascertain accrual rates—we will progress to the main trial if we succeed in recruiting ≥12 practices per month (expected recruitment is ≥21 practices/month for the main trial). Incorporated in the trial are (1) a process evaluation investigating determinants of GPs' referral behaviour and the implementation of the intervention in practice and (2) a health economic evaluation in a child's lifetime.

### Eligibility and recruitment

GP practices registered in England that carry out 6-week hip checks (at age 42–76 days) and agree (1) to being randomised and (2) to hospitals releasing data concerning infants they examine in the trial. Ineligible are practices planning to close with 12 months of trial start.

We anticipate that ≥110 infants per year will undergo the 6-week hip check in each of the recruited practices. Each practice will recruit infants for a period of 15 months and infants will be followed for 2 years. This trial is planned therefore to run for 44 months across 152 GP practices.

## Retention

The proposed trial is not onerous on the practice staff and practices will not be challenged by excessive study-related manoeuvres. Because the primary outcome will be collected using practice and hospital databases only, it is likely that we will attain high levels of follow-up. We will send periodic newsletters to GPs and foster regular contacts.

## Randomisation

Practices will submit an eligibility form to the clinical trials unit, which will register and randomly allocate (concealed) practices within 48 hours. An independent statistician will coordinate randomisation with an allocation ratio of 1:1 and stratified by practice size (based on observed sizes obtained from expressions of interest—we expect two or three strata).

## Interventions
### Control intervention

GPs will assess infant hips following best practice principles; they will be provide a leaflet about the national best practice recommendations.[2]

### Experimental intervention

This is a complex intervention comprising a video developed specifically for this trial and a diagnostic aid in form of a 9-item binary checklist. At the time when informed consent is taken, GPs will watch the video. They will then examine all infants in the 6-week check according to best practice principles, but with the addition of the diagnostic aid (implemented electronically to each practice's computer system).

According to national guidelines, the 6-week check shall capture infants at age 6–10 weeks and infants in whom a diagnosis of DDH is considered should see a specialist within 2 weeks.[2] Participating GPs of both intervention arms will be asked to adhere to this policy.

## Definitions for the purpose of this trial

'Appropriate referral' denotes a referred hip deemed 'clinically significant', that is, is treated or monitored (at least one follow-up) by a specialist surgeon. Any ambiguous cases will be reviewed by our expert advisory panel who will assign an ultimate diagnosis in consensus. 'Clinically insignificant' are referred hips resulting in reassurance and discharge from surgeons' clinics (ie, no treatment or monitoring). 'Late diagnoses' are cases of DDH diagnosed by a specialist surgeon at age 3–24 months.[15 16]

## Sample size calculation

We based its calculation on Poisson regression and audit data from the Nottingham area (partially presented in Price et al[7]) and University College Hospital (audit of 3 years' practice, unpublished data): a conservative estimate of the number of infants referred as a consequence of the 6-week check within the recruitment period of 1 year is, on average, 3 per practice. Based on the same data,

we estimate that about one in three (or, on average, one per practice) of such referrals is correctly made ('appropriate' referrals, defined above as 'clinically significant', are a subset of these 'correct' referrals). Thus, on average, we expect 2 'incorrect' referrals per practice. These are the ones we seek to reduce, and the intervention should serve to cut them by 50%. Thus we aim to detect a reduction, on average, from 3 to 2 referrals per practice in the intervention arm. To account for correlation within practices, which can also be thought of as overdispersion relative to the underlying Poisson variation, we assumed a between-practice component of variance of 20% of the average Poisson 'counting' variance per practice. For 90% statistical power and testing two sided at p=0.05, we need 76 practices per group (15% of this total comprises a safety margin to allow for potential challenges). Average referral rates from the Nottingham (2.8%) and University College Hospital data (2.7%) are very consistent, and suggest a target of about 110 children per practice per year to undergo the 6-week check.

## Primary trial endpoint

Number of referred infants that are considered 'clinically insignificant'—this is a measure of the clinical importance of referrals in the 2-week hip pathway. We chose this outcome because it is extremely relevant clinically and one for which it is possible to power the trial.[2] An intervention could be successful at achieving this outcome while missing infants who do have DDH; we thus specified a principal secondary endpoint. False negatives are also important but, because of their rarity, would not be an efficient endpoint on which to power the trial (and would thus be outside International Council for Harmonisation of Technical Requirements for Registration of Pharmaceuticals for Human Use (ICH) E9),[17] which recommends that the primary outcome should be both clinically and statistically convincing). A research assistant will retrieve these data from respective hospital electronic systems by conducting site visits to all secondary care facilities to which GPs refer infants, starting 6 weeks after the last patient enters the trial.

## Secondary trial endpoints
### Infant level

► Number of appropriate referrals per practice (principal secondary endpoint): for all infants referred as a consequence of the 6-week check, a researcher will collect, using unique NHS patient numbers, and categorise the appropriateness of referrals by a practice with respective hospital databases using a standardised taxonomy, blind to practice random allocation. She/he will conduct site visits to all such hospitals starting 6 weeks after the last patient entered the trial.
► Number of late diagnoses: We will employ deterministic methods of data linkage. Using unique NHS numbers of any recruited infant, we will obtain from Health and Social Care Information Centre the corresponding Health Episodes Statistic (HES)

identifiers. HES is a data warehouse that includes all hospital admissions and outpatients' visits occurring in all hospitals in England. In HES, we will establish whether any infant in the trial was admitted to a UK hospital as a result of DDH using relevant International Statistical Classification of Diseases and Related Health Problems (ICD)-10 codes and Office of Population Censuses and Surveys (OPCS)-4 codes.[18] We will extract, for the whole trial period, the full hospital history for all infants in the trial (in and out patient episodes), collate all episodes into a combined exploratory analysis view, collate interactions, and allocate outcomes to practice groups to facilitate outcome analyses.[19] These data will be compared with the data collected by the researcher (see above); the use of these two strategies will enhance the robustness of data.

► Consequences of late diagnoses: Using data collected above we will record nature and frequency of such consequences (nature of treatment, length of hospital stay, frequency of secondary care contacts).
► Health-related quality of life (parental proxy report): Child-Health-Utility-9D.[20]

### Process level
► Volume of referrals: Total number of patients referred to secondary care during the trial period. This variable will be collected prospectively at practice level by the participating GP and forwarded monthly to the clinical trial unit.
► Timeliness of referrals: In infants referred to secondary care, we will measure the days from referral issued to hospital appointment (collected in the same way as primary endpoint). This process measure will inform about target wait times[2] achieved.

### Clinician level
► Confidence and attitudes of GP and secondary care clinicians towards the diagnostic aid: will be assessed 12 weeks after trial set-up and at trial completion using a modified measure based on the theory of planned behaviour.[21]
► Implementation issues and acceptability of diagnostic aid among GPs and secondary care providers: we will conduct qualitative research at the trial end to elicit this information.
► Use of diagnostic aid and acceptability of intervention: a self-administered questionnaire to evaluate the use of the aid will be posted to all the GPs in the intervention group; also direct observations and interviews will ascertain this outcome.

### Parent/carer level (collected from one in ten parents/carers)
► General worry: State-Trait Anxiety Inventory 6-items short form.[22]
► DDH-related worry: Infant Hip Worries Inventory.[23]
► Satisfaction with trial: dimensions of care items from EUROPEP[24] (table 1).

### Statistical analysis
#### Primary endpoint
We will compare the randomised practices using Poisson mixed models, accounting for extra Poissonian variability by including random intercept terms for practices. The response variable will be the number of clinically insignificant referrals from each practice with an offset in the linear model of the log(e) total number of children checked in each practice, to account for differences in practice size and constitution. The random effect at the practice level will account for overdispersion.

#### Principal secondary endpoint

#### Late diagnoses
As we anticipate small numbers of events in each randomised comparison we may not be in a position to account for practices using random intercept terms. Where this is the case we will report the total numbers over a minimum 2-year observation and compare the overall group scores using Fisher's exact test.

#### Quantitative outcome measures
We will summarise scores by instrument, accounting for practices and report difference in group means for each treatment arm. We expect mean scores to be lower in the intervention arm for State-Trait Anxiety Inventory and Infant Hip Worries Inventory, but higher or equal for EUROPEP.

#### Missing data
For the primary endpoint and principal secondary endpoint the data collection methods should identify qualifying episodes. Because of the nature of these data the conventional concept of missingness does not directly apply (eg, we will not have individually randomised subjects who cannot be followed up). However, if a practice withdraws from the trial, we will explore the consequences of this action by assuming a poor outcome among that practice if in the intervention group and a good outcome if in the control condition, to identify the potential consequences of their withdrawal. Complete case analyses will be conducted for secondary outcomes. If there is a mismatch between practices in the two treatment conditions, we will consider undertaking joint models examining simultaneously the binomial of missingness and the outcome measure of interest.

### Health economics
#### Cost analysis
We will analyse the cost associated with the intervention compared with usual practice for the entire trial period, and examine costs from the perspective of the NHS and of families. The cost components included in main analysis are: cost of 6 weeks in both intervention arms; any subsequent referrals, diagnostic tests and treatment. We will collect costs about GP time, which we will multiply by unit costs from routine sources.[25]

**Table 1** Assessments at different time points

| | Baseline | At 6 week hip check | Within 2 weeks of intervention | 3 month follow-up | 15 month follow-up | 2 year follow-up |
|---|---|---|---|---|---|---|
| **Clinician-level** | | | | | | |
| Theoretical Domains Framework questionnaire | X | | X | X | X | X |
| GP characteristics questionnaire | X | | | | | |
| Fidelity questionnaire | | | X | X | X | X |
| Non-participant observations | | X | | | | |
| Semi-structured interviews | | | X | | X | |
| **Parent/carer-level** | | | | | | |
| State Trait Anxiety Inventory six item short form | | | X | | | |
| Infant Hip Worries Inventory | | | X | | | |
| EUROPEP | | | X | | | |
| Non-participant observations | | X | | | | |
| Semi-structured interviews | | | X | | | |
| Out-of-pocket-costs questionnaire | | | X | | | |
| EuroQol 5 Dimension Scale (EQ-5D) | | | X | | | X |
| Child Health Utility 9D (CHU 9D) | | | | | | X |
| Willingness-to-pay questionnaire | | | | | X | |

GP, general practitioner.

## Within-trial cost-effectiveness analysis

With the costs described above we will produce a dataset of patient-level within-trial costs and outcomes. We will calculate the incremental cost per clinically insignificant referral avoided and the incremental cost per late diagnosis avoided. Using bootstrapping of the mean cost and outcomes differences, we will estimate confidence intervals around the incremental cost-effectiveness ratios.[26] With the bootstrap replications, we will construct a cost-effectiveness acceptability curve to show the probability that the aid is cost-effective for different values of NHS willingness to pay for outcomes. We will perform deterministic sensitivity analyses.

## Long run cost–utility analysis

We will use several measures to evaluate the lifetime cost-effectiveness of the intervention. We will ask 20 carers of infants aged 2–4 years to complete (1) on behalf of their children the Child-Health-utility-9D[20] and (2) for themselves the EQ-5D-5L,[27] both measure health-related quality of life.

## Cost–benefit analysis

With data from the trial about the impact of the intervention on appropriate referrals, we will calculate the monetary value that parents place on the intervention using willingness-to-pay methodology.[28] This will provide an estimate of the monetary value of the additional benefits (positive/negative) of the intervention. We will calculate the net benefit of the intervention by subtracting the incremental cost of the aid, as calculated above from the trial data, from the monetary value of the additional benefit. Following recruitment of the last infant in the trial, we will recruit 200 carers of infants undergoing the 6-week check from trial-participating practices. These will be 100 carers whose infants will be referred to secondary care as a consequence of the 6-week check and 100 who will not. They will complete a self-report questionnaire that utilised several techniques[28] to elicit willingness-to-pay values. We will calculate willingness-to-pay values for the whole sample and test for variations based on socio-demographic groups and referral to secondary care.

## Integrated qualitative and quantitative process evaluation

This workstream will explore the implementation, adherence to protocol, receipt and setting of the intervention. We will examine the views of all groups of participants on the intervention; study how the intervention is implemented; investigate contextual factors that affect the intervention; and study how effects vary in subgroups of GPs. These data will help in understanding how, for whom and why the trial had effects and the extent to which outcomes result from issues of trial fidelity and implementation. We will collect process data from all 152 sites including clinician and carer outcomes. We will conduct alongside the trial non-participant observation and semi-structured interviews. We will include a purposive sample of 10 practices for the qualitative study, interviewing 4–5 participants in each (eg, GP, carers, hospital consultant), resulting in 40–50 interviews. This sample will include a small number of practices in the control arm (for comparative purposes), and a range of intervention practices to include different locations, practice sizes and types. We will analyse process data before outcome data to avoid bias in interpretation.[29] Interviews will be audio recorded and transcribed, data from observations will be recorded contemporaneously using a template. Data from process outcomes, interviews (transcripts) and observations will be analysed from the perspective of both behavioural change theory[30] and normalisation process theory.[31]

## Strategies to mitigate potential bias

Since this effectiveness trial will test whether the intervention can work under usual circumstances, we will rely on paediatric orthopaedic surgeons in determining the ultimate diagnosis of DDH. Variations in the surgeons' diagnostic accuracy are inevitable hence the need for a randomised study. We will perform analyses by surgeon (or hospital) to quantify this variation. Blinding of GPs, practice staff, carers is impossible; however, most such outcomes will be assessed with validated questionnaires. Primary and principal secondary endpoints will be collected by an independent researcher blinded to treatment allocation. In case an infant is referred to hip ultrasound without an orthopaedic consultation, a trial-appointed advisory panel shall review the scan blinded and according to standard methods[32] to avoid reporting bias. There is a risk for verification bias—while our trial includes a 2-year follow-up to capture late presenting DDH, we cannot rule out that some infants with DDH will remain undiagnosed within this period, thus underestimating the number of late diagnosed DDH. However, the 2-year mark has previously been found to be a robust outcome.[33]

## Patient and public involvement

We developed this protocol with carers of children with DDH and the founding director of 'Steps', a charity supporting patients with lower limb disorders. We discussed the need for the trial and trial procedures and conduct with staff members of GP practices. Our established patient and public involvement group has reviewed and commented on this protocol and will periodically review, support and advise on the conduct of the trial.

## Trial and data management

The trial will be run through PRIMENT Clinical Trials Unit and conducted in accordance with established quality management systems and standardised operating procedures (online supplemental appendix 1). All data will be handled in accordance with the UK Data Protection Act 2018. All analyses will be conducted blinded to allocation groups.

## Ethics and dissemination

Leicester Central Research Ethics Committee (19/ EM/0317, 16 January 2020) and Health Research

Authority Confidentiality Advisory Group (19/CAG/0198, 18 February 2020) approved protocol V.2. 'Section 251' approval was obtained, which omits the need for written informed consent from parents/carers; consent will be obtained at cluster level from the lead GPs. We will publish results in peer-reviewed journals and disseminate results to patient organisations and the media.

## DISCUSSION

This trial is part of a programme of research to improve the diagnosis of DDH: consensus-based diagnostic criteria were established in prior research, tailored for use in primary care, supplemented by a video designed for GPs. There has only been one randomised trial on the topic of DDH in the UK[33] but it explored the use of ultrasound screening—our trial focusses on the compulsory '6-week check'. Because the intervention tested in this trial is based on consensus of clinical experts, there is a risk that the opinions of experts change as clinical knowledge evolves. However, the criteria of the diagnostic aid have been in use for decades and will likely not loose relevance in the foreseeable future. The collection of outcome data from various hospitals connected to GP practices will be challenging; use of national health services databases should mitigate this challenge. While our trial includes a 2-year follow-up period to capture late presenting DDH, we cannot rule out that some infants with DDH will remain undiagnosed within this period. This trial has the potential to improve the compulsory 6-week hip check with use of a relatively simple intervention. It will also provide an understanding of the cost-effectiveness of the intervention in a whole lifetime horizon of a 6-week old. If successful, the intervention can be rolled out to clinical services relatively easily and at low costs.

**Contributors** AR, IN and NF were involved in conception and trial design. JG expanded and developed the qualitative process evaluation of the protocol. AC designed the behavioural change psychology aspects of the trial, working closely with JG and AR. RH, AR and IN developed the health economic evaluation plan. JW, IN, AR and IN developed all statistical aspects of the trial. AR and KW drafted the manuscript for publication. IN, JG, AC, IN, JW, NF, RH, KW and AR critically revised the article for important intellectual content and approved the final version of the manuscript. AR is chief investigator of the trial and together with IN, NF, JW, JG and AC secured its funding. We acknowledge the valuable input from Professor Daniel Ray in relation to the collaborative work with NHS Digital for outcome data collection.

**Funding** This paper presents independent research funded by the National Institute for Health Research (NIHR) under its Programme Grant for Applied Research funding stream (RP-PG-0616-20006).

**Disclaimer** The views expressed are those of the authors and not necessarily those of the NIHR or the Department of Health and Social Care.

**Competing interests** None declared.

**Patient consent for publication** Not required.

**Provenance and peer review** Not commissioned; externally peer reviewed.

**ORCID iDs**
Andreas Roposch http://orcid.org/0000-0002-0143-7840
Rachael Hunter http://orcid.org/0000-0002-7447-8934

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
