## [Reviewer comments · BMJ Open]

ARTICLE DETAILS

TITLE (PROVISIONAL)	Study protocol for evaluation of aid to diagnosis for developmental dysplasia of the hip in general practice: controlled trial randomised by practice
AUTHORS	Roposch, Andreas; warsame, kaltuun; Chater, Angel; Green, Judith; Hunter, Rachael; Wood, J; Freemantle, Nick; Nazareth, Irwin

VERSION 1 – REVIEW

REVIEWER	Sattar Alshryda Al Jalila Children's Speciality Hospital. Dubai UAE
REVIEW RETURNED	20-Jul-2020

GENERAL COMMENTS	Thank you very much for asking me to review the research protocol entitled “Evaluation of aid to diagnosis for developmental dysplasia of the hip in general practice: controlled trial randomised by practice.” The authors have proposed a study protocol to evaluate a newly developed standardised diagnostic criteria for DDH. These criteria are based on international Delphi consensus and consist of a 9-item checklist that the authors believe it has the potential to enable non-experts to identify children with DDH with more accuracy. Authors stated that there is no current tool to aid examination or offer guidelines for referral. Authors designed a randomised controlled trial by General Practice (GP) comparing the newly developed diagnostic aid against standard care for the infant hip check. The primary objective is to determine whether this aid (the 9-item check list) to the diagnosis of DDH reduces: 1. The number of clinically insignificant referrals from primary care to hospital.2. The number of late diagnosed DDH. The trial would also include a qualitative process evaluation, an assessment of professional behaviour change and a full health economic evaluation.
--

Authors planned to recruit 152 GP practices in England. These would be randomised to conduct the hip checks with use of the study diagnostic aid and/or as per usual practice.

The total number of infants seen during a 15-month recruitment period would be 110 per practice (total 16,720). Two years after the 6-week hip check, authors would measure the number of referred infants that are (1) clinically insignificant for DDH and (2) those that constitute appropriate referrals.

Introduction:

The authors nicely described the background of the condition (DDH) that they are investigating and the current challenges and shortfalls. Clinical examination by general practitioners (and even by paediatric orthopaedic experts) is not sensitive enough to detect DDH particularly hip dysplasia without dislocation which resulted in either delayed diagnosis or unnecessary referral fearing delayed diagnosis.

Authors recently developed criteria based on expert's consensus to aid diagnosis. (1, 2). They also stated that they validated their diagnosis criteria (3). I could not find the quoted validation study and I appreciate if this could be shared with me. Validation of a diagnostic clinical criteria is a very complex process. Did the quoted validation study cover the various aspects of validity such as construct validity, content validity, criterion validity, predictive validity and concurrent validity?

They stated that diagnosis criteria demonstrated validity in a study of 44 patients referred from GPs to hospital, have a positive predictive value (PPV) of 89% (95% confidence interval 70, 97) and a negative predictive value (NPV) of 76% (95% confidence interval, 50-96).

Authors refined the criteria to a checklist of 9 items, for use by GPs during the 6-week hip check-up and developed a training video, featuring a GP. The video explains the difference between the Barlow and Ortolani manoeuvres, how to test for a leg length inequality, and how to identify limitations in hip abduction. Some of these manoeuvres such as Barlow's and Ortolani tests requires manual appreciation and they are difficult (if not impossible) to learn from watching a training video.

As a reader and a practitioner, I would like to know why the researchers decided to use their criteria instead of hip ultrasound (the golden standard).

The ultrasound is readily available, cheap, safe, and it has become an extended part of clinical examination rather than an investigation. Ultrasound has a better diagnostic profile for the condition (sensitivity of 88.5% (95% confidence interval 84.1% to 92.1%), specificity of 96.7% (96.4% to 97.4%), a positive

predictive value of 61.6% and a negative predictive value of 99.4%.) (4). The International Interdisciplinary Consensus Committee on DDH Evaluation (ICODE) has recently published a consensus paper in which they strongly recommended using ultrasound to evaluate infants' hip abnormality as the most accurate mean and they cautioned against using clinical methods only.(5)

Having trained 100s of professionals of various backgrounds (doctors, sonographers, physiotherapists, nurse practitioners) to perform hip ultrasound to detect DDH, I believe it is easier to train general practitioners to perform a hip ultrasound with reasonable accuracy rather than train them to evaluate a baby hip clinically.

The authors need to discuss and expand on their choice of the intervention. Would they consider adding a third group (US)?

Objectives

Three objectives are listed:

1. To determine whether an aid to the diagnosis of DDH reduces the number of clinically insignificant referrals from primary care to hospital, and the number of late diagnosed DDH.
2. To determine the cost-effectiveness of this intervention.
3. To conduct an integrated qualitative and quantitative process evaluation in order to understand all participants' experience in the trial and with the intervention; study how the intervention is implemented; investigate contextual factors that affect the intervention.

They are reasonable and the design allows to calculate them.

METHODS AND ANALYSIS

Design:

Practice randomisation seems to be reasonable. However, the process of recruitments and retention requires more thoughts. Several factors reduce the willingness of GPs to participate in trials. (6)

1. If the research question is not of sufficient interest or not relevant to general practice, then GPs are unlikely to take part.
2. The process of obtaining informed consent can also be an important barrier to GP participation as some may worry that taking part in research could disrupt the "normal" doctor-patient relationship.
3. Lack of financial incentives may also deter GP involvement.
4. Lack of time.

The rate of recruitment of 21 practice per month (nearly one a day if we exclude weekend) may be optimistic.

Eligibility and recruitment

Eligibility criteria are reasonable and numbers are realistic.

Retention

See design.

Randomisation

More elaboration about concealment is required. Whom would be concealed and how to ensure concealment with future interaction during training, data collection, medical records inspections.

Interventions

The intervention is clearly described. I expressed earlier my reservation on using any intervention whether for diagnosis or treatment for research purpose when there is a proven superior alternative.

It was not clear who would sign the informed consent! I understood, the GP would sign the consent form which is appropriate.

However, a few ethical considerations need clarification:

1. Would parents be informed about the study? If yes, there must be a patient / parents information sheet to complement the study protocol.
2. If not, how would you ethically justify involving children in a study without parents knowing that they are involved in

a study. I hope authors would consider what we learnt from the Alder Hey affair (7)

Definitions for the purpose of this trial

The definitions are clear. Some thoughts are required about the followings issues:

1. Some GP practice have access for US machine or service and this will make most of their referral appropriate in contrast to GP practices with no access to US machine.
2. Adding a layer of validation (Expert Advisory Panel) is not appropriate for two reasons:
 - a. It violates the intention to treat principle. As per the protocol, the definition of primary outcome is decided by the specialist surgeon (treated, monitored or discharge).
 - b. It creates ethical dilemma when the experts panel disagree with the specialist surgeons. What would they do if a child was not received the correct treatment by a specialist surgeon? Would they ignore? Would they interfere?

Sample size calculation

It was difficult to follow how the calculation was made and whether the assumptions are correct or even realistic. They anticipated that there would be 3 referrals per GP practice per year. They estimated that about 1 in 3 (or, on average, 1 per practice) of GP referrals to hospitals is correctly made ('appropriate' referrals'). Thus, on average, they expected 2 'incorrect' referrals per practice.

Authors based their calculation on two sources of data. One is published by Price et al.(8). She stated "*Between July 1990 and June 2005, there were 112 084 live births within the old Nottingham Area Health Authority. Of these children, 13 491 (12.0%) were referred for assessment of the hip and 455 (0.4%) received some sort of treatment.*" If I understood the protocol's definition correctly, the appropriate referrals should be around 0.4% and not 30%!

I think authors need to clarify how these figures were estimated.

Primary and secondary trial endpoint

Both primary and secondary outcome measures are reasonable and clinically significant.

I think the following statement (line 38 page 8) should have read “.....An intervention could be successful at achieving this outcome ~~while~~ without missing infants who do have DDH; we thus specified a principal secondary end....”

The researchers try to get as much as outcomes and sideways studies as possible and they should be careful this may inhibit recruitment and retention.

Statistical analysis & Health economics

I am not familiar with some of the statistical tests that authors had planned and a statistician’s and health economist’s opinion would be useful here.

The rest of the protocol

No reservation

Conclusions and recommendations

In conclusion, authors have invested enormous amount of time and efforts and I congratulate them on reaching this far in setting such an ambitious study. I wish it was a different intervention, namely hip ultrasound instead of the 9-item check list. However, I have no doubt conducting this study will add value to our understanding of the condition and the current practice.

Given the protocol have been approved by NIHR and received grant to conduct the study, I would support its publication. Would the outcome of this research change my practice? I doubt it very much. Hip ultrasound enables me to see what inside the baby hip. Clinical examination does not; therefore, hip US would remain my first choice.

Once again, thanks for asking me to review this study.

Sattar Alshryda

Consultant Trauma and Paediatric Orthopaedic Surgeon

	References  1. Roposch A, Protopapa E, Cortina-Borja M. Weighted diagnostic criteria for developmental dysplasia of the hip. The Journal of pediatrics. 2014;165(6):1236-40.e1. 2. Roposch A, Liu LQ, Hefti F, Clarke NM, Wedge JH. Standardized diagnostic criteria for developmental dysplasia of the hip in early infancy. Clin Orthop Relat Res. 2011;469(12):3451-61. 3. Roposch A, E. P. Testing newly developed standardized diagnostic criteria for DDH in consecutive patients referred to a paediatric orthopaedic unit. Pediatric Orthopedic Society of North America. 2013. 4. Woolacott NF, Puhan MA, Steurer J, Kleijnen J. Ultrasonography in screening for developmental dysplasia of the hip in newborns: systematic review. BMJ. 2005;330(7505):1413. 5. O'Beirne JG, Chlapoutakis K, Alshryda S, Aydingoz U, Baumann T, Casini C, et al. International Interdisciplinary Consensus Meeting on the Evaluation of Developmental Dysplasia of the Hip. Ultraschall in der Medizin (Stuttgart, Germany : 1980). 2019;40(4):454-64. 6. Bell-Syer SEM, Thorpe LN, Thomas K, MacPherson H. GP Participation and Recruitment of Patients to RCTs: Lessons from Trials of Acupuncture and Exercise for Low Back Pain in Primary Care. Evidence-Based Complementary and Alternative Medicine. 2011;2011:687349. 7. Bauchner H, Vinci R. What have we learnt from the Alder Hey affair? That monitoring physicians' performance is necessary to ensure good practice. BMJ. 2001;322(7282):309-10. 8. Price KR, Dove R, Hunter JB. Current screening recommendations for developmental dysplasia of the hip may lead to an increase in open reduction. The bone & joint journal.95-B(6):846-50.
--	--

REVIEWER	Alexander Aarvold Southampton Children's Hospital, UK University of Southampton, UK
REVIEW RETURNED	21-Jul-2020

GENERAL COMMENTS	Highly important question. Topical. Relevant nationally and internationally. Good primary outcomes - ie numbers of late detected (missed) cases and numbers of 'unnecessary' referrals. The study is a thorough trial of the 6-week baby check, ie part of the national screening process. That should be made clear in the title. From the title it is unclear if the study refers to accurate detection of kids of walking age with hip dysplasia ie those missed by the current screening process. The 6 week check is a contentious topic, highly relevant to GP's, and this study would have more impact if this was included in the title. I have concern that the numbers of late detected cases will be unreliable. This is due to:  • small numbers if only 110 infants per year are included, introducing likely Type 1 & 2 errors • Movement of population out of region. Thus late detected cases 2 years later would not appear on local hospital records
---

	• Many late detected cases occur after 2 years of age. See Broadhurst et al 2019 BJJ Without robust data on late detected cases, then any health economic evaluation may not be accurate. This should be highlighted in the limitations. And the economic analysis is likely only to be relevant for the costs of referral rather than surgical costs for the late detected cases. References to include: Broadhurst et al 2019, as above. This highlights the importance of the question being asked. Plus include reference(s) on the cost of medical negligence claims (Page 3, Line 59).
--	---

REVIEWER	Takeshi Toyooka Nishikawa Orthopaedic Clinic, Japan
REVIEW RETURNED	09-Aug-2020

GENERAL COMMENTS	Thank you for your great contribution. Cause of DDH are rare, Poisson mixed models and Fishers exact test are appropriate for this study. In the Quantitative outcome measures, I don't know why you summarize and transform for some questionnaires. Does it mean combine each score? I think it's better to analyze them separately. Please describe what kind of analysis will you use.
--

REVIEWER	WILLIAM Timothy BROX, MD UCSF Fresno Fresno, CA, USA
REVIEW RETURNED	11-Aug-2020

GENERAL COMMENTS	This is a robust study. I look forward to hearing about the results of the trial.
---

REVIEWER	Kim Madden McMaster University, Canada
REVIEW RETURNED	14-Aug-2020

GENERAL COMMENTS	Thank you for this interesting protocol for a cluster randomized trial. I have only a few comments because I think this paper is well done. Really excellent identification of the problem and potential impact of the study. I can immediately see the importance.  1. More info on the stratification would be helpful. E.g. how many strata? What is the cutoff? 2. Will there be a formal interim analysis? Are there formal stopping rules? I see that the steering committee will function as the safety monitoring committee. Will there be someone independent of the trial to ensure participant safety?
--

VERSION 1 – AUTHOR RESPONSE

Reviewer: 1

I could not find the quoted validation study and I appreciate if this could be shared with me.

We were (are) unable to publish this study because of the risk to contaminate the trial. A publication will require to discuss our tool in detail. The tool could then be discovered and taken up by professionals for use in clinical practice, and our trial jeopardised. The validation study was funded by the same agency, NIHR, who has obtained a report before we progressed to the trial. We thus would appreciate if this could be accepted in this way.

Did the quoted validation study cover the various aspects of validity such as construct validity, content validity, criterion validity, predictive validity and concurrent validity?

As we validated a tool that is a diagnostic aid, for the purpose of our aims it was relevant to concurrent criterion validity, as well as content and face validity. There is nothing predictive in our work because this is not the aim of our tool; neither was there a construct to be tested and validated (we tested criterion validity).

They stated that diagnosis criteria demonstrated validity in a study of 44 patients referred from GPs to hospital, have a positive predictive value (PPV) of 89% (95% confidence interval 70, 97) and a negative predictive value (NPV) of 76% (95% confidence interval, 50-96).

That is correct. This was published in a POSNA Conference Abstract book, which is reference 11 in our manuscript.

Some of these manouvres such as Barlow's and Ortolini tests requires manual appreciation and they are difficult (if not impossible) to learn from watching a training video.

The video is not meant to teach professionals how to examine a baby hip. That is training they have already undertaken when they became GPs. As we stated, the video is part of a complex intervention and its aim is to introduce the diagnostic aid. Part of this introduction is a demonstration of the items of the aid. (We are not doing a study whereby we like to re-train professionals, instead our study provides a diagnostic aid for professionals who have all undergone – more or less – the very same training; and who examine baby hips as part of their daily work.)

As a reader and a practitioner, I would like to know why the researchers decided to use their criteria instead of hip ultrasound (the golden standard).

We refer to our 'Rationale' section and our previous research on the matter. Our study's aim was to improve the 6-week hip check, which, by its nature, is compulsory consultation by a GP without the use of any imaging tests. The consultation will, in turn, conclude the need for ultrasound testing in some cases. Our aim was to investigate what happens in this consultation and thereafter.

Having trained 100s of professionals of various backgrounds (doctors, sonographers, physiotherapists, nurse practitioners) to perform hip ultrasound to detect DDH, I believe it is easier to train general practitioners to perform a hip ultrasound with reasonable accuracy rather than train them to evaluate a baby hip clinically.

This may be another study one could design and conduct. Our study is of a very different nature, rationale, and scope though. It has been funded with GBP 2M which is perhaps evidence that our rationale and scope is justified in light of the current evidence base.

The rate of recruitment of 21 practice per month (nearly one a day if we exclude weekend) may be optimistic.

This number is based on careful work we did to plan recruitment, numbers, etc. It applies to the local setting in England, where this trial will be taking place. NIHR CRNs have been consulted in estimating the figures. The figures are further based on similar trials we did.

More elaboration about concealment is required. Whom would be concealed and how to ensure concealment with future interaction during training, data collection, medical records inspections.

One a cluster has signed up for the trial and consent given, the CTU will tell them immediately in which treatment arm they are because otherwise the cluster cannot set up their infrastructure (a software will need to be installed for those in the experimental treatment arm) as required for the trial. We randomise clusters of GPs within a cluster all will be the same.

Would parents be informed about the study? If yes, there must be a patient / parents information sheet to complement the study protocol.

Parents will be informed about the trial and there is an approved information leaflet. However, BMJ Open guidelines do not state that such documents should be included in the manuscript. We can provide this if required for publication; however, it is not clear to us if publication of such documents would be justified given the sensitive nature of the trial that is about to start. Ethical approval has been given to include children without the need for written informed consent. We made reference to this in the section “Ethics and dissemination”.

Some GP practice have access for US machine or service and this will make most of their referral appropriate in contrast to GP practices with no access to US machine.

We are certain that no GP has ultrasound machines in their surgery to use for the 6-week hip check. This is not the case in the UK and we know this from our focus groups. They have access to referring to ultrasound imaging in secondary care but this is part of our trial pathway, and we have accounted for variability in referral modalities. Given the limited word count allowed for the manuscript, all such details could not be provided in the manuscript.

Adding a layer of validation (Expert Advisory Panel) is not appropriate for two reasons: It violates the intention to treat principle. As per the protocol, the definition of primary outcome is decided by the specialist surgeon (treated, monitored or discharge). It creates ethical dilemma when the experts panel disagree with the specialist surgeons. What would they do if a child was not received the correct treatment by a specialist surgeon? Would they ignore? Would they interfere?

The panel will review images after the trial has finished. We had written under the section “Strategies to mitigate for potential bias” the following:

“In case an infant is referred to hip ultrasound without an orthopaedic consultation, a trial-appointed advisory panel shall review the scan blinded and according to standard methods³² to avoid reporting bias.”

Authors based their calculation on two sources of data. One is published by Price et al.(8). She stated “Between July 1990 and June 2005, there were 112 084 live births within the old Nottingham Area Health Authority. Of these children, 13 491 (12.0%) were referred for assessment of the hip and 455 (0.4%) received some sort of treatment.” If I understood the protocol’s definition correctly, the appropriate referrals should be around 0.4% and not 30%! I think authors need to clarify how these figures were estimated.

We stated that the data we used to estimate sample size is partly discussed and presented in Price et al. Available to us was more data from this same area that we used to estimate the sample size. We have a long document on this subject that we provided to the funding body but we are not able to add all this information as this would be well beyond the space limits. We hope that both reviewer and editor will give us the benefit of a doubt that our text reflects the figures we worked with. These were indeed those summarised to a very condensed space in this manuscript for BMJ Open.

Both primary and secondary outcome measures are reasonable and clinically significant. I think the following statement (line 38 page 8) should have read “.....An intervention could be successful at achieving this outcome whilst without missing infants who do have DDH; we thus specified a principal secondary end....”

The sentence is correct as we wrote it with “whilst”. This word justifies the use of a principle secondary endpoint. It is clear in our view, and was so for all reviewers here (and those who reviewed on behalf of the funder, NIHR).

I wish it was a different intervention, namely hip ultrasound instead of the 9-item check list.

That may be so, but we feel that the need for research currently, and in our country, is where we are going to set the focus: the compulsory 6-week check, done as a compulsory assessment of any newborn in the UK.

Would the outcome of this research change my practice? I doubt it very much. Hip ultrasound enables me to see what inside the baby hip. Clinical examination does not; therefore, hip US would remain my first choice.

The main participants of our trial are GPs in the UK. If the results are positive, the trial has the potential to change the practice of GPs (not surgeons or other professionals in the UK).

Reviewer: 2

I have concern that the numbers of late detected cases will be unreliable. This is due to: small numbers if only 110 infants per year are included, introducing likely Type 1 & 2 errors; Movement of population out of region. Thus late detected cases 2 years later would not appear on local hospital records. Many late detected cases occur after 2 years of age.

With 152 clusters and n=110 infants annually, we estimate to include 16,000 to 17,000 six-week-olds that we will follow, with use of NHS Digital, for 2 years.

Under the section “strategies to mitigate for potential bias” we had written

“There is a risk for verification bias – while our trial includes a 2-year follow up to capture late presenting DDH, we cannot rule out that some infants with DDH will remain undiagnosed within this period, thus underestimating the number of late diagnosed DDH. However, the 2-year mark has previously been found to be a robust outcome.³³”

We thus believe we have addressed the issue of false negatives as a consequence of followup period.

On request of the reviewer, we have added this comment once more in Discussion as a potential limitation.

“While our trial includes a 2-year followup period to capture late presenting DDH, we cannot rule out that some infants with DDH will remain undiagnosed within this period.”

Without robust data on late detected cases, then any health economic evaluation may not be accurate. This should be highlighted in the limitations.

We have developed a decision analysis model to infer all relevant estimates. As usual for such models, sensitivity analyses will address the aspect mentioned by the reviewer. We thus don't quite see that this is a limitation because models only ever can provide estimates with a confidence interval (and that we will do).

And the economic analysis is likely only to be relevant for the costs of referral rather than surgical costs for the late detected cases.

Obviously, our health economic model is one for diagnostic tests. It thus includes all relevant costs, including those for surgery. As we mentioned in the manuscript, it captures a child's life time horizon.

Plus include reference(s) on the cost of medical negligence claims (Page 3, Line 59).

We are unable to provide a published reference and have deleted the statement of the magnitude of the financial loss for the NHS through litigation.

Reviewer: 3

In the Quantitative outcome measures, I don't know why you summarize and transform for some questionnaires. Does it mean combine each score? I think it's better to analyse them separately. Please describe what kind of analysis you will use.

We will summarise scores by instrument, i.e. for each outcome measure we will provide its score separately. Transformation refers to taking steps if, for example, any scores are skewed by group.

Reviewer: 5

More info on the stratification would be helpful. E.g. how many strata? What is the cutoff?

We expect there to be 2 or 3 strata, this will depend on the nature of the actual clusters to take part in the trial. We expect there being small (up to three GPs for a practice) and large (more than three GP for a practice). We added this information.

Will there be a formal interim analysis? Are there formal stopping rules?

Yes, but in line with other similar publications in BMJ Open and the limited word count allowed, we have omitted such details. Instead we focused on the main aspects of analysis.

I see that the steering committee will function as the safety monitoring committee. Will there be someone independent of the trial to ensure participant safety?

The steering committee is independent of the trial with 3 independent members. All such members have no connection with the CTU; they are not involved in the trial.

VERSION 2 – REVIEW

REVIEWER	Takeshi Toyooka Nishikawa Orthopaedic Clinic
REVIEW RETURNED	27-Aug-2020

GENERAL COMMENTS	The statistics has been significantly improved so it's easier to understand.
--

REVIEWER	Kim Madden McMaster University, Canada
REVIEW RETURNED	14-Sep-2020

GENERAL COMMENTS	I would prefer if the authors added brief details about the formal interim analysis and stopping rules for transparency but I will leave this to the editor to decide if it is necessary. Otherwise, the authors have addressed all of my comments.
---

VERSION 2 – AUTHOR RESPONSE

Reviewer comments:

The statistics has been significantly improved so it's easier to understand.

No changes required.

I would prefer if the authors added brief details about the formal interim analysis and stopping rules for transparency but I will leave this to the editor to decide if it is necessary. Otherwise, the authors have addressed all of my comments.

We are not planning interim analysis and there are no stopping rules. We have specifically written this out (page 12 – statistical analysis section)